# Effects of multiple sclerosis on the audio-vestibular system: a systematic review

Evrim Gür,[1] Ghada Binkhamis  ,[1,2] Karolina Kluk  [1]

EG and GB contributed equally.

[1]Manchester Centre for Audiology and Deafness (ManCAD), School of Health Sciences, Ellen Wilkinson Building, The University of Manchester, Manchester, UK
[2]Communication and Swallowing Disorders, King Fahad Medical City, Riyadh, Saudi Arabia

**Correspondence to**
Dr Ghada Binkhamis;
gbinkhamis@gmail.com

## ABSTRACT

**Objective** Systematically investigate the effects of multiple sclerosis (MS) on the audio-vestibular system.
**Methods** Systematic review of literature investigating audio-vestibular conditions in persons with MS (PwMS) aged ≥18 years. PubMed, Scopus, NICE and Web of Science were searched. Randomised controlled trials, and cohort, case–control, observational and retrospective studies in English, published from 2000 to 21 November 2021, evaluated PwMS with at least one outcome (pure tone audiometry, auditory brainstem response, otoacoustic emissions, cortical auditory evoked potentials, functional MRI assessing auditory function, vestibular evoked myogenic potentials, videonystagmography, electronystagmography, posturography, rotary chair, gaps in noise, word discrimination scores, duration pattern sequence test), were included. Study selection and assessments of bias were independently conducted by two reviewers using the Risk of Bias Assessment Tool for Non-randomized Studies, Newcastle-Ottawa Scale (NOS) and the NOS adapted for cross-sectional studies.
**Results** 35 studies were included. Auditory function was evaluated in 714 PwMS and 501 controls, vestibular function was evaluated in 682 PwMS and 446 controls. Peripheral auditory function results were contradictory between studies; some found abnormalities in PwMS, and others found no differences. Tests of brainstem and central auditory functions were more consistently found to be abnormal in PwMS. Most vestibular tests were reported as abnormal in PwMS, abnormalities were either peripheral or central or both. However, quantitative analyses could not be performed due to discrepancies between studies in results reporting, test stimulus and recording parameters.
**Conclusions** Although abnormal results on auditory and vestibular tests were noted in PwMS, specific effects of MS on the audio-vestibular system could not be determined due to the heterogeneity between studies that restricted the ability to conduct any quantitative analyses. Further research with consistent reporting, consistent stimulus and consistent recording parameters is needed in order to quantify the effects of MS on the auditory and vestibular systems.
**PROSPERO registration number** CRD42020180094.

## INTRODUCTION

Multiple sclerosis (MS) is a chronic, immune-mediated, inflammatory disease that causes a neurodegenerative process in the central nervous system (CNS).[1] There are four main types of MS: relapsing remitting (RRMS), primary progressive (PPMS), secondary progressive (SPMS) and progressive relapsing (PRMS). The autoimmune process of MS results in CNS plaques (focal areas of demyelination) in addition to axonal injury or loss.[2 3] These plaques most commonly occur in the white matter of the brain, cerebral cortex including subpial regions, spinal cord and optic nerve,[4] resulting in optic neuritis, double vision, tremor, ataxic gait, weakness and numbness in one or more limbs.[5] It has been suggested that demyelination plaques can also occur in the central auditory and vestibular pathways resulting in hearing loss and balance disorders.[6] For example, plaques may occur in the brainstem, where both efferent and afferent auditory and vestibular pathways are present.[6] Also, the vestibular nuclei and the root entry zone of the eighth cranial nerve have been shown to be one of the most common neuroanatomic locations for inflammatory demyelination[7]; in addition, peripheral neural connections in the internal auditory canal and within the inner ear structures may be affected by demyelination resulting in peripheral auditory and vestibular involvement in persons with MS (PwMS).[8–10] The extent of occurrence of audio-vestibular symptoms in PwMS is unclear, as reports on their prevalence vary in the literature. For example, one study reported a prevalence of audio-vestibular symptoms between 1% and

## STRENGTHS AND LIMITATIONS OF THIS STUDY

⇒ This study provides a comprehensive systematic review of the literature on multiple sclerosis (MS) and the auditory and vestibular systems from 2000 until 21 November 2021.
⇒ This is the only systematic review that investigated the effects of MS on the audio-vestibular system.
⇒ Differences between studies in reporting of results and in test procedures restricted the ability to conduct any form of quantitative analyses.

$28\%^{[11]}$; whereas others reported that vertigo is seen in 20%–35% of PwMS, and hearing loss is seen in 1%–17% of PwMS,[12–15] while a recent systematic review on MRI and auditory tests found that 25% of PwMS had either sudden or progressive hearing loss.[11]

Literature on the involvement of the auditory system in PwMS is either contradictory or limited. First, several studies that assessed the effects of MS on the peripheral auditory system through evaluating hearing thresholds via pure tone audiometry (PTA) found worse PTA thresholds in PwMS compared with healthy controls,[16–18] while one study did not find any significant differences in PTA thresholds between PwMS and healthy controls.[19] Second, some studies that assessed the effects of MS on the brainstem auditory pathway via auditory brainstem responses (ABR) found that PwMS had abnormal ABRs,[11 20] while one study found no differences in ABRs between PwMS and healthy controls.[21] Finally, only a few studies investigated the effects of MS on speech understanding as assessed via word discrimination scores (WDS) and on temporal processing as assessed via the gaps in noise (GIN) test. These found poorer WDS in noise and poorer performance on the GIN in PwMS compared with healthy controls.[22 23]

Literature on the involvement of the vestibular system in PwMS generally found abnormalities in PwMS. Studies that assessed the vestibular system via electronystagmography (ENG) found abnormal results in PwMS on tests of peripheral vestibular function, central vestibular function or both.[24–27] Other studies that investigated the vestibular pathway via vestibular evoked myogenic potentials (VEMPs) in PwMS found abnormalities in VEMP results including latency delays, reduced amplitudes and interaural side differences.[28–39] Further studies investigated the functional integrity of the vestibular system in PwMS via static and dynamic posturography, and found that PwMS performed in general worse than healthy controls.[40–42] Although literature has shown abnormalities on different tests of vestibular function in PwMS, different studies used different test procedures, different parameters, or reported different components of each test result (eg, VEMP latency vs amplitude).

In summary, numerous studies performed different audio-vestibular test batteries in PwMS. These studies aimed to uncover possible MS-related changes in auditory or vestibular function. However, results from literature that investigated auditory function varied, as some indicated no differences, while others reported considerable differences between PwMS and healthy controls. Whereas results from literature on the effects of MS on vestibular function generally showed abnormalities in PwMS, the effects of MS on central versus peripheral vestibular function are still unclear. Therefore, literature needs to be combined and summarised to better understand the potential effects of MS on the auditory and/or vestibular systems. If auditory and/or vestibular disorders in PwMS are better understood, early identification and early intervention could occur, resulting in improved quality of life. This highlights the need for this systematic review with a focus on the effects of MS on the audio-vestibular system. We hypothesise that PwMS experience auditory and vestibular disorders at a higher rate than the general population. This systematic review aims to investigate this hypothesis by examining and synthesising current literature on the effects of MS on the audio-vestibular system.

## METHODS
This systematic review followed the Preferred Reporting Items for Systematic Reviews and Meta-Analyses statement,[43] and was pre-registered in PROSPERO (CRD42020180094).

### Patient and public involvement
No patient was involved.

### Data sources and searches
A systematic search was conducted in the electronic bibliographic databases PubMed, NICE, Scopus and Web of Science using the following terms: 'Multiple Sclerosis', 'Auditory Function', 'Vestibular Function', 'Hearing Loss', 'Dizziness', 'Vertigo'. Operators (AND, OR) were used to narrow the search (see online supplemental file for detailed search strategy). A total of 21 searches were performed by EG until 21 November 2021. No language or date restrictions were applied.

### Inclusion and exclusion criteria
Studies were eligible for inclusion if they met the following criteria: (1) investigated audio-vestibular conditions; (2) participants were diagnosed with any type of MS (RRMS, SPMS, PPMS, PRMS); (3) participants were aged 18 years and older because of maturational effects on outcomes of some audio-vestibular tests; (4) one or more of the preselected tests described below were used to evaluate auditory or vestibular function; (5) any of the following study types: randomised controlled trials, or cohort, case–control, observational or retrospective studies; (6) results from PwMS were either compared with a healthy control group or with normative data; (7) any geographical location. Studies were excluded based on the following: (1) did not include the keywords 'Multiple Sclerosis', 'dizziness', 'auditory function', 'vestibular', 'vertigo' or 'hearing loss' in the title or abstract; (2) no available full text in English; (3) published before the year 2000; (4) participants were diagnosed with comorbid diseases.

### Audio-vestibular tests
Evaluation of auditory function was restricted to the following measures: PTA, ABR, otoacoustic emissions (OAEs: transient evoked (TEOAE) and distortion product (DPOAE)), cortical auditory evoked potentials (CAEP) and functional MRI assessing auditory function. Evaluation of vestibular function was restricted to the following measures: VEMPs including cervical (cVEMPs)

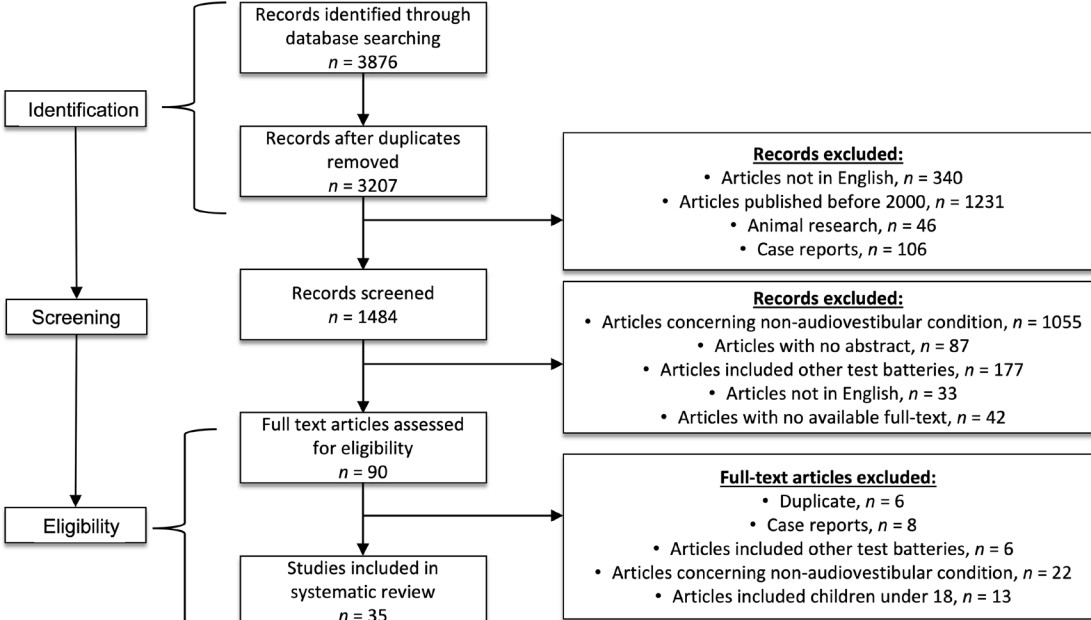

**Figure 1** Flow diagram of systematic literature review.

and ocular (oVEMPs), videonystagmography (VNG), ENG, posturography and rotary chair. Above tests were defined as primary outcomes; the secondary outcomes were defined as measures of speech perception and temporal processing, specifically: GIN test, WDS and duration pattern sequence test (DPST). (Refer to online supplemental file for details on which measures and parameters were extracted for each test.)

## Study selection

EG manually removed the duplicates and screened the titles and abstracts for inclusion and exclusion criteria. The remaining full texts were independently reviewed by EG and GB, and any disagreements were resolved by discussion or by involving KK until a consensus was reached.

## Assessment of bias

Risk of bias assessment was done independently by EG and GB. Case–control studies were assessed using the Risk of Bias Assessment Tool for Non-randomized Studies (RoBANS[44]) to examine the selection of participants, confounding variables, measurement of exposure, blinding of outcome assessment, incomplete outcome of data and selective outcome reporting. Cohort studies were assessed with the Newcastle-Ottawa Scale (NOS[45]). A judgement of 'low risk' of bias, 'high risk' of bias or 'unclear risk' of bias was provided for each domain. Cross-sectional studies were evaluated with the NOS adapted for cross-sectional studies.[46] Any disagreements were resolved by discussion or by involving the third reviewer (KK) until a consensus was reached.

## Data extraction and synthesis

The following data were extracted when available: (1) study details: aims and design; (2) population: recruitment methods; diagnosis and inclusion/exclusion criteria; syndrome differentiation; course of disease; number of

participants: screened, eligible, enrolled, included in the analysis; age; gender; (3) details of the pathology: specific lesion site, primary or secondary, stable or progressive, additional handicaps related to MS; (4) methodology: test procedure, stimulus and recording parameters, electrode and interelectrode impedance levels; (5) test results (eg, PTA thresholds, VEMP amplitudes, ABR peak latencies, etc).

EG extracted the data and GB confirmed the accuracy of all extracted data. Any disagreements were resolved by discussion between the two reviewers. Per pre-registered protocol, investigators were not contacted for unreported data or additional information due to time constraints and given current COVID-19 pandemic-related global circumstances.

## Criteria for data synthesis

Studies that used the same test procedure (either PTA, ABR or VEMP), minimum five participants in each study, minimum two studies for each test method and the same stimulus and recording parameters were used. Mean, SD and/or individual values were extracted when available; however, the effect size could not be calculated due to inconsistencies between studies in stimulus and recording parameters or in how results were reported.

## RESULTS
### Description of included studies

A total of 3876 studies were identified initially, and 1484 studies were identified after duplicate removal and primary limitations (figure 1); 1055 were excluded because they did not relate to audio-vestibular conditions, 42 due to no full text availability, 33 were not in English, 177 due to the use of tests other than those in the inclusion criteria and 87 were excluded because they did not have an abstract. The remaining 90 studies were

examined for inclusion, 8 were excluded because they were case reports, 22 did not evaluate audio-vestibular conditions, 6 used tests other than those in the inclusion criteria, 13 included participants under 18 years of age and 6 additional duplicates were removed.

A total of 35 studies underwent full-length review. Fourteen evaluated the auditory system, 18 evaluated the vestibular system and three evaluated both auditory and vestibular systems.

### Assessment of bias

Case–control studies assessed with RoBANS (online supplemental table S1) were judged as low risk of bias in most categories, except for blinding of outcome assessment that was judged as high risk of bias in all 22 studies. Cohort studies assessed with NOS (online supplemental table S2) were judged as either good (n=2), poor (n=2) or unknown (n=2, rating could not be assigned due to one or more categories that were judged as 'not applicable'; eg, two studies were evaluated as not applicable for 'selection of the non-exposed group' due to lack of a control group). Cross-sectional studies were assessed using NOS adapted for cross-sectional (online supplemental table S3) and most were judged as good (n=6) with one judged as poor.

### MS and auditory function

Seventeen studies investigated the auditory system involvement in PwMS (table 1—see online supplemental table S4 for an alternative table format summarising findings based on outcome): 3 assessed OAEs, 5 assessed PTA, 13 evaluated ABRs, 2 investigated CAEPs and 3 of the 17 studies performed speech perception and/or temporal processing tests. No studies that conducted functional MRI to assess auditory function were located.

Three studies compared OAEs in PwMS and healthy controls with a total number of 218 participants: 120 PwMS and 98 controls. Two studies did not find differences between PwMS and controls in TEOAEs and/or DPOAEs.[14 47] However, the third study found weaker responses in both TEOAEs and DPOAEs in PwMS compared with healthy controls.[21]

Five studies performed PTA on PwMS to assess hearing levels. A total of 510 participants (265 PwMS and 245 healthy controls) were evaluated with PTA, sample size varied from 80 to 146 and age range was between 18 and 65 years. Two out of five studies reported no significant difference in PTA thresholds between PwMS and controls.[19 21] One reported worse PTA thresholds only in female PwMS compared with female controls,[13] and two reported worse thresholds in PwMS compared with controls,[18 48] with one finding worse thresholds in participants with SPMS when compared with RRMS.[18] Three of five studies provided mean and SD values of PTA thresholds (online supplemental table S5). However, reported values could not be pooled because the way that authors presented the data differed between each other. For instance, Doty *et al* reported average PTA thresholds across right and left

ears for each frequency in male and female participants separately,[19] while Di Mauro *et al* reported values for each ear separately for all participants.[21]

Thirteen studies assessed ABRs (table 1) and included 881 participants (548 PwMS and 333 healthy controls), sample size varied from 25 to 126 and age range was between 18 and 68 years. Twelve of the 13 studies reported abnormalities in different ABR components in PwMS as follows:
- ► Prolonged absolute latencies of wave I,[47] wave III and wave V.[47–50]
- ► Prolonged interpeak latencies of I–III,[14 28 30 48–53] III–V[14 28 30 48 50–53] and I–V.[20 28 47–51 53]
- ► Interaural interpeak latency differences for I–III and I–V[28] and III–V.[28 51]
- ► Absent waves III and V.[28 30 48 51 52]
- ► Absent waves at high stimulus presentation rates.[54]
- ► Reduced wave V amplitude.[53]
- ► Increased I/III and I/V amplitude ratios.[51]
- ► Reduced V/I amplitude ratio.[48 50]
- ► Poor waveform morphology and/or only wave I within normal limits.[20]

Only Di Mauro *et al* did not find any differences in ABRs between PwMS and controls.[21] Six out of 13 studies shared mean and SD values for latencies, amplitudes and/or interaural latency differences (online supplemental table S6); however, data could not be combined, and the effect size calculation was not performed due to the variability in stimulus and recording parameters. For example, Matas *et al* presented rarefaction clicks at 80 dB normal hearing level (nHL) with a presentation rate of 19.9 clicks per second,[49] while Kaytancı *et al* presented clicks (polarity unspecified) at 70 dB nHL with a presentation rate of 13.0 clicks per second.[47]

Cortical evoked potentials were evaluated by two studies using two different tests (table 1). Japaridze *et al* reported 30% abnormality in slow cortical potentials; abnormalities were either absent peaks or prolonged peak latencies.[51] Matas *et al* assessed P300 peak latency (defined as a cognitive potential) and did not find any significant differences between PwMS and healthy controls; however, percentage of abnormalities in P300 results was higher in PwMS (16%) compared with controls (0%).[49]

Three studies assessed speech perception and temporal processing (table 1) with a total of 188 participants: 93 PwMS and 95 healthy controls and an age range of 18–65 years. Two of the three studies found that PwMS had poorer WDS in noise and an increased approximate threshold plus lower per cent correct answers on the GIN test compared with controls[22 23]; and one of the two studies also found that PwMS had lower per cent correct answers on the DPST.[22] However, the third study found no significant difference between PwMS and controls on WDS in quiet.[18]

### MS and vestibular function

A total of 21 studies that investigated the vestibular system in PwMS were identified (table 2—see online

**Table 1** Characteristics of studies examining the effects of MS on auditory function

| Author (year) | Study type | Sample size | | PwMS mean age (years) | PwMS age range (years) | % female | MS type (sample size) | MS duration mean (range) years | Test battery | Results of PwMS compared with controls/normative data |
|---|---|---|---|---|---|---|---|---|---|---|
| | | PwMS | Healthy controls | | | | | | | |
| Japaridze et al (2002)[51] | Case–control | 40 | 33 | 30.5 | 18–57 | 77.5 | ? | ? | ABR / CAEP (SCP) | 65% abnormal / 30% abnormal |
| Versino et al (2002)[28] | Cohort | 65 | 18 | 35.5 | 19–61 | ? | ? | ? | ABR | 37.5% abnormal |
| Burina et al (2008)[52] | Cohort | 60 | NA | 37.2 | ? | 68.3 | RRMS | ? | ABR | 95% abnormal |
| Eleftheriadou et al (2009)[30] | Cohort | 46 | 40 | 40 | 20–66 | 45.6 | RRMS | 4.6 | ABR | 26% abnormal |
| Lima et al (2009)[20] | Case–control | 25 | NA | 42.6 (female) 38 (male) | 33–53 (female) 24–56 (male) | 64.0 | RRMS (14), PPMS (8), unspecified progressive (3) | ? | ABR | 30% abnormal |
| Lewis et al (2010)[18] | Case–control | 47 | 49 | 51.4 | 21–65 | 57.7 RRMS, 28.6 SPMS | RRMS (26), SPMS (21) | RRMS: 12.6 (2–43) SPMS: 23.3 (8–50) | PTA / WDS | Worse thresholds / No significant difference |
| Matas et al (2010)[49] | Case–control | 25 | 25 | 34.88 | 25–55 | 76.0 | RRMS | 4.25 (?) | ABR / CAEP (P300) | Significantly different / 16% abnormal |
| Doty et al (2012)[19] | Case–control | 73 | 73 | Males: 45.24 Females: 45.6 | ? | 71.2 | RRMS (57), PPMS (3), SPMS (6), unspecified (7) | Males: 7.36 (?) Females: 7.84 (?) | PTA | No significant difference |
| Saberi et al (2012)[14] | Case–control | 60 | 38 | 29.9 | ? | 73.3 | ? | 3.2 (?) | PTA / TEOAE and DPOAE / ABR | Worse thresholds in female PwMS / No significant difference / 20% abnormal |
| Valadbeigi et al (2014)[22] | Case–control | 26 | 26 | 28.9 | 18–40 | ? | RRMS | ? | GIN / WDS / DPST | Significantly different / Lower scores PwMS / Significantly different |
| Pokryszko-Dragan et al (2015)[53] | Case–control | 86 | 40 | 39.55 | 19–60 | 72.0 | ? | 8.57 (1–30) | ABR | Significantly different |
| Kaytancı et al (2016)[47] | Cohort | 20 | 20 | 31.3 | ? | 55.0 | ? | ? | TEOAE / ABR | No significant difference / Significantly different |
| Di Mauro et al (2019)[21] | Cohort | 40 | 40 | 37 | 18–50 | 60.0 | RRMS | 0.8 (0.8–2) | PTA / TEOAE and DPOAE / ABR | No significant difference / Significantly lower / No difference |

Continued

**Table 1** Continued

| Author (year) | Study type | Sample size PwMS | Sample size Healthy controls | PwMS mean age (years) | PwMS age range (years) | % female | MS type (sample size) | MS duration mean (range) years | Test battery | Results of PwMS compared with controls/normative data |
|---|---|---|---|---|---|---|---|---|---|---|
| Elbeltagy et al (2019)[23] | Case-control | 20 | 20 | 37.6 | 30–50 | ? | RRMS | ? | GIN | Significantly different |
|  |  |  |  |  |  |  |  |  | WDS | Significantly lower |
| Delphi et al (2021)[50] | Cross-sectional | 25 | 25 | 31.43 | 18–45 | 72.0 | ? | ? | ABR | Significantly different |
| Rishiq et al (2021)[54] | Case-control | 11 | 9 | 49.5 | 34–68 | 72.7 | RRMS (9), PPMS (1), SPMS (1) | 11.9 (2–30) | ABR | Significantly different at high click rate |
| Srinivasan et al (2021)[48] | Case-control | 45 | 45 | 31.77 | 18–50 | 73.3 | RRMS | 6.1 (5.37) | PTA | Significantly different |
|  |  |  |  |  |  |  |  |  | ABR | Significantly different and 55.56% abnormal |

Cells shaded in grey represent test results that were abnormal or showed a difference between PwMS and healthy controls.

?, not reported; ABR, auditory brainstem response; CAEP, cortical auditory evoked potential; DPOAE, distortion product otoacoustic emission; DPST, duration pattern sequence test; GIN, gaps in noise; MS, multiple sclerosis; NA, not applicable; PPMS, primary progressive multiple sclerosis; PTA, pure tone audiometry; PwMS, persons with multiple sclerosis; RRMS, relapsing remitting multiple sclerosis; SCP, slow cortical potential; SPMS, secondary progressive multiple sclerosis; TEOAE, transient evoked otoacoustic emission; WDS, word discrimination scores.

supplemental table S7 for an alternative table format summarising findings based on outcome). Two studies investigated vestibular function using ENG; 3 used static posturography, 1 used dynamic posturography, 14 used cVEMPs and 4 also used oVEMPs, and 2 used rotary chair testing.

A total of 90 participants (60 PwMS and 30 healthy controls) were evaluated using ENG, age ranged from 24 to 64. Both studies found ENG abnormalities in the peripheral and central vestibular systems in PwMS.[27 55] Degirmenci et al noted that 90% of PwMS showed ENG abnormalities compared with 6.7% of the controls, and there was a significant difference between groups.[27] Abnormalities suggestive of central vestibular pathology were found in 83.3% of PwMS and in 6.7% of controls, and abnormalities suggestive of peripheral vestibular pathology were found in 36.7% of PwMS and in 0% of controls.[27] Zeigelboim et al also reported ENG abnormalities in 86.7% (83.4% peripheral) of their PwMS.[55]

Four studies assessed posturography with a total of 314 participants (187 PwMS and 127 healthy controls). All four studies found significant differences between PwMS and controls.[40 41 56 57] Two studies reported static posturography results for the same sample,[40 41] they noted increased delineated area, average sway and speed of sway for both open and closed eyes in PwMS compared with controls. A third study also conducted static posturography and found that sway area and path length were significantly greater in PwMS than in controls.[56] The fourth study evaluated dynamic posturography and found longer motor control test latencies, lower equilibrium scores on the sensory organisation tests and a compromised ability to use three sensory inputs (somatosensory, vestibular and visual preference) in PwMS compared with controls.[57]

A total of 14 studies assessed VEMPs and included 724 participants (435 PwMS and 289 healthy controls), sample size varied from 30 to 88 and age range was between 18 and 71 years. Four studies evaluated both oVEMPs and cVEMPs[35 58–60] and 10 studies evaluated cVEMPs.[28–30 50 61–66] Thirteen of the 14 studies reported abnormalities in different VEMP components in PwMS as follows:

► Prolonged latencies of p13,[28–30 35 50 58 60–66] n23[28 30 35 50 58 60–66] and p1 and n1.[35 58 60]
► Reduced p13-n23 amplitude.[28 50 61 63]
► Interside asymmetry for p13,[28 29 63] and n23 and n23 latencies.[28]
► Absent waves.[30 62 63 66]

Ten of 14 studies compared VEMPs in PwMS to healthy controls.[30 35 50 58–63 65] Except for two,[59 65] all found a significant difference between PwMS and controls. The remaining VEMP studies used data from controls as normative data.[28 29 64 66]

Thirteen studies shared mean and SD values of p13, n23, p1 and n1 latencies (online supplemental tables S8 and S9). However, data could not be analysed because intensity level, stimulus type, stimulus polarity and other

**Table 2** Characteristics of studies examining the effects of MS on vestibular function

| Author (year) | Study type | Sample size | | PwMS mean age (years) | PwMS age range (years) | % female | MS type (sample size) | MS duration mean (range) years | Test battery | Results of PwMS compared with controls/ normative data |
| --- | --- | --- | --- | --- | --- | --- | --- | --- | --- | --- |
| | | PwMS | Healthy controls | | | | | | | |
| Sartucci and Logi (2002)[61] | Case–control | 15 | 15 | 44.5 | 26–59 | 66.6 | ? | ? | cVEMP | Significantly different |
| Versino et al (2002)[28] | Cohort | 70 | 18 | 35.5 | 19–61 | ? | ? | ? | cVEMP | 31.4% abnormal |
| Alpini et al (2004)[29] | Case–control | 40 | 25 | 38 | 17–71 | 57.5 | ? | ? | cVEMP | 70% abnormal |
| Aidar and Suzuki (2005)[62] | Case–control | 15 | 15 | 39.3 | ? | ? | ? | ? | cVEMP | Significantly different |
| Patkó et al (2007)[63] | Case–control | 30 | 30 | 43.4 | 27–60 | 66.6 | ? | ? | cVEMP | Significantly different |
| Zeigelboim et al (2008)[55] | Case–control | 30 | 0 | 42.23 | 27–64 | 80 | RRMS | ? | ENG | 86.7% abnormal |
| Eleftheriadou et al (2009)[30] | Cohort | 46 | 40 | 40 | 20–66 | 45.6 | RRMS | 4.6 (?) | cVEMP | Significantly different |
| Degirmenci et al (2010)[27] | Cohort | 30 | 30 | 37.9 | 23–56 | 56.7 | RRMS | ? | ENG | 90% abnormal |
| Gabelić et al (2013)[58] | Case–control | 30 | 15 | ? | ? | 46.6 | RRMS | 3.93 (0.2–21) | cVEMP / oVEMP | Significantly different / Significantly different |
| Harirchian et al (2013)[64] | Case–control | 20 | 20 | 30 | 20–40 | 50 | RRMS SPMS | ? | cVEMP | 70% abnormal |
| Parsa, et al (2015)[35] | Cross-sectional | 34 | 15 | 29.8 | ? | 100 | ? | ? | cVEMP / oVEMP | Significantly different / Significantly different |
| Doty et al (2018)[57] | Case–control | 58 | 72 | Males: 44.61 Females: 44.60 | ? | 68.9 | ? | Male: 7.03 (?) Female: 6.54 (?) | Dynamic posturography | Significantly different |
| Kavasoğlu et al (2018)[65] | Case–control | 30 | 31 | 30 | 18–45 | 60.0 | ? | <1 (?) | cVEMP | 23.3% abnormal / No significant difference |
| Koura and Hussein (2018)[66] | Case–control | 20 | 10 | 36.80 | ? | 65.0 | ? | 4.4 (?) | cVEMP | 100% abnormal |
| Inojosa et al (2020)[40 41] | Cross-sectional | 99 | 30 | 35.01 | 18–50 | 68.7 | ? | 5.5 (?) | Static posturography | Significantly different |
| Yang and Liu (2020)[56] | Cross-sectional | 30 | 25 | 50.8 | ? | 76.7 | ? | 14 (?) | Static posturography | Significantly different |
| Delphi et al (2021)[50] | Cross-sectional | 25 | 25 | 31.43 | 18–45 | 72.0 | ? | ? | cVEMP | Significantly different |
| Elmoazen et al (2021)[60] | Case–control | 20 | 10 | Brainstem lesions: 40 No brainstem lesion: 34.1 | ? | ? | ? | ? | cVEMP / oVEMP | Significantly different / Significantly different |
| Cochrane et al (2021)[59 67] | Cross-sectional | 40 | 20 | 42.4 | 21–55 | 88.0 | RRMS | 9.9 | Rotary chair / cVEMP / oVEMP | Significantly different / No significant difference / No significant difference |

Cells shaded in grey represent test results that were abnormal or showed a difference between PwMS and healthy controls.
?, not reported; cVEMP, cervical vestibular evoked myogenic potential; ENG, electronystagmography; MS, multiple sclerosis; oVEMP, ocular vestibular evoked myogenic potential; PwMS, persons with multiple sclerosis; RRMS, relapsing remitting multiple sclerosis; SPMS, secondary progressive multiple sclerosis.

VEMP recording parameters differed between studies. For instance, Sartucci and Logi presented clicks at 140 dB sound pressure level at a rate of 3 per second,[61] Aidar and Suzuki presented clicks at 95 dB hearing level (HL) at a rate of 2 per second,[62] Kavasoğlu *et al* presented clicks at 100 dB nHL at a rate of 5 per second[65] and Koura and Hussein presented a 500 Hz tone burst at 95 dB HL.[66]

Two studies conducted rotary chair testing. However, they tested and reported results from the same sample.[59 67] Forty PwMS and 20 healthy controls were tested, results showed that PwMS had lower vestibular-ocular reflex cancellation gain with a visual target and larger variances in their responses on both the subjective visual vertical and horizontal tests compared with controls.[59] Also, that PwMS performed worse than controls on central vestibular measures but not on peripheral.[67]

## DISCUSSION

In this systematic review, 35 studies were reviewed: 22 case–control studies, 6 cohort studies and 7 cross-sectional studies. The aim was to investigate the effects of MS on the audio-vestibular system by examining and synthesising current literature that used different auditory and vestibular tests. Additionally, this systematic review presents an overview of the last 21 years of literature specific to MS in the context of the auditory and vestibular systems. The selected outcome measures for the auditory system were OAE, PTA, ABR, CAEP, WDS, GIN and DPST. Vestibular system outcome measures were VNG, ENG, rotary chair, VEMPs and static and dynamic posturography. The included studies on auditory function evaluated 1215 individuals in total (714 PwMS and 501 healthy controls), the included studies on vestibular function covered 1128 individuals (682 PwMS and 446 healthy controls). Included studies assessed participants who were over the age of 18, and the geographical location of individuals was not limited to a specific continent, country or hemisphere.

Neither meta-analyses nor effect size calculations could be performed due to the inconsistencies across studies in reporting results or in test stimulus and recording parameters that lead to the inability to combine their data (online supplemental tables S5, S6, S8, and S9). This is a significant methodological limitation especially that the impact of this systematic review on clinical practice would have been much greater had quantitative analyses been performed.

### Auditory system and MS

The specific effects of MS on the auditory system, based on the results of auditory tests included in this systematic review, could not be established.

### OAEs in PwMS

The effect of MS on the peripheral auditory system, as demonstrated by OAEs, was only found by one of the three included studies.[21] Di Mauro *et al* reported abnormal OAEs with normal ABRs in participants newly diagnosed with RRMS, showing cochlear rather than retrocochlear involvement.[21] They suggested that this cochlear dysfunction may be caused by a demyelinating process in the brainstem's medial olivocochlear bunch that innervates outer hair cells[68] resulting in decreased OAE amplitudes.[21] However, reduced OAE amplitudes were not reported by others, and it is unclear why Di Mauro *et al*[21] found differences between PwMS and controls, and both Kaytancı *et al* and Saberi *et al* did not.[14 47] Therefore, there is very limited evidence to date on the effects of MS on OAEs, and there is a need for further investigations.

### PTA in PwMS

Effects of MS on PTA thresholds could not be defined. Three of five included studies reported worse PTA thresholds in PwMS compared with controls.[14 18 48] However, Srinivasan *et al* included only participants with normal hearing, that is, test sample was biased towards better PTA[48]; therefore, the difference between PwMS and controls, although statistically significant, is not clinically significant. Apart from Di Mauro *et al*, who found no difference between PwMS and controls but only included participants with normal hearing who were recently diagnosed with RRMS,[21] reasons behind differences in PTA results between studies are unclear. Type and duration of MS do not appear to be factors, as those who found a difference included participants with a mixture of MS types and MS durations. For example, Lewis *et al* included participants (n=47) with RRMS (mean MS duration: 12.6 years) and SPMS (mean MS duration: 23.3 years),[18] and Saberi *et al* included PwMS (n=60, MS type not reported) with a mean MS duration of 3.2 years,[14] and they both found differences in PTA thresholds between PwMS and controls, while Doty *et al* did not find any difference in PTA thresholds with the inclusion of participants (n=73) with different MS types (RRMS, PPMS, SPMS, unspecified MS) and a mean MS duration of 7.36 years in males and 7.84 years in females.[19] Therefore, conclusions on the effects of MS on PTA thresholds cannot be reached given the limited literature and the inability to combine results for a meta-analysis. Thus, there is a clear need for further studies investigating changes in PTA thresholds in PwMS.

### ABRs in PwMS

The effects of MS on the brainstem auditory pathway as demonstrated by ABRs were more evident; 12 of 13 included studies found abnormalities in different ABR components in PwMS.[14 20 28 30 47–54] Demyelination plaques tend to occur in the brainstem, and the impairment of the auditory olivocochlear pathways by demyelination is possible.[6] Therefore, ABR abnormalities such as prolonged latencies are thought to be related to the effects of MS plaques in the brainstem on the integrity of the auditory pathways.[11] Abnormal ABR findings in PwMS reported by studies included in this review were consistent with previous literature.[25 69] Only Di Mauro *et al* found no differences in ABRs between PwMS and controls;

however, PwMS with hearing loss or brainstem lesions in their MRI were excluded from the study, plus the authors included only participants newly diagnosed with RRMS.[21] Matas *et al* included normally hearing participants with RRMS (mean MS duration: 4.25 years) and found differences in ABRs between PwMS and controls.[49] Burina *et al* also included participants with RRMS and brainstem or cerebellum lesions and found abnormal ABRs.[52] Therefore, the reasons Di Mauro *et al*[21] did not find differences in ABRs between PwMS and controls are likely related to the short duration of MS and the exclusion of participants with brainstem lesions. Although most studies that investigated ABRs in PwMS found abnormalities, the nature of these abnormalities could not be established. This is due to the differences between studies in reported ABR measures (eg, absolute peak latencies vs peak amplitudes) and in stimulus and recording parameters (eg, presentation levels, stimulus polarity, transducer). Therefore, there is a clear need for consistency across studies to better understand the effects of MS on ABRs.

### CAEPs in PwMS

The effects of MS plaques on the integrity of the central auditory pathway as measured by CAEPs were much less studied; only two studies were included in this systematic review.[49 51] Both found abnormalities suggesting possible cortical involvement in PwMS; however, these studies could not be compared, and data could not be synthesised due to the use of different testing techniques. Therefore, further studies are needed to better understand the effects of MS on CAEPs.

### Speech perception and temporal processing in PwMS

The effects of MS on speech perception and temporal processing were also infrequently studied; three studies were included in this systematic review.[18 22 23] Two of the three studies found abnormalities on WDS tests in PwMS, both studies that found abnormalities performed WDS testing in noise,[22 23] and the study that did not find abnormal WDS tested in quiet.[18] It is therefore likely that Lewis *et al* did not find abnormal WDS in PwMS because they did not test in background noise.[18] This is supported by Valadbeigi *et al*'s findings of worse WDS in noise than in quiet in PwMS.[22] These abnormalities in WDS are possibly related to MS lesions in the central auditory pathways. Any degeneration caused by MS plaques could lead to a misperception of the acoustic signal leading to poorer performance on WDS.[70] Additionally, the two studies that performed the GIN test found abnormalities in PwMS indicating lower temporal resolution performance than in controls.[22 23] A previous study suggested that persons with lesions in the central auditory system had worse temporal resolution performance and therefore poorer performance on the GIN test compared with healthy controls.[71] Therefore, it is possible that temporal resolution is affected by MS plaques in the CNS. Since the evidence regarding abnormalities in speech perception and temporal processing in PwMS was limited, this systematic review concludes that there is a need for further research to evaluate MS effect on functional auditory performance.

### Vestibular system and MS

All studies included in this systematic review found abnormalities on vestibular tests suggesting that the vestibular system may be more affected by MS than the auditory system.

### ENGs in PwMS

The two included studies that performed ENGs in PwMS found abnormalities suggestive of peripheral and/or central vestibular pathology.[27 55] Zeigelboim *et al* primarily found peripheral vestibular pathology (83.4%) in their participants with RRMS,[55] while Degirmenci *et al* mainly found central vestibular pathology (83.3%) in their participants with RRMS.[27] These contradictory results might be due to the inclusion of participants with RRMS in both studies, as persons with RRMS may differ in terms of lesion site in the CNS, severity of impairment, developmental stage of disease and frequency of attacks.[55] Given this limited and contradictory evidence, the effects of MS on ENGs are still unknown. Therefore, more literature is needed on different MS subtypes to better understand this.

### Posturography in PwMS

The four included studies that investigated postural control using either static or dynamic posturography found abnormal posturography results in PwMS.[40 41 56 57] To maintain balance, the CNS evaluates the input from the vestibular, visual and somatosensory systems, then weighs the input according to where reliable information is coming from, and MS may affect each of these three systems.[56] Maintaining body balance in PwMS requires increasing the weight on the other two systems if one sensory system input is impaired. Yang and Liu concluded that PwMS rely mostly on cues from visual and proprioceptive systems to maintain their balance.[56] Lord *et al* and Doty *et al* also stated that PwMS had difficulty maintaining their balance when visual feedback and proprioceptive feedback were inconsistent or conflicting.[57 72] In addition, a previous study reported that PwMS have slower somatosensory conduction than healthy individuals.[73] Thus, PwMS might receive impaired or delayed proprioceptive feedback resulting in postural instability.[56 73 74] Another possible cause of instability in PwMS is that motor neurons might be affected by demyelination plaques, leading to an impairment in the transmission from the CNS to the motor end units.[56 75] Although all four included studies found greater instability and impaired balance in PwMS, the evidence regarding the effects of MS on functional balance outcomes is very limited. Further research is required to make a clear conclusion on this subject.

### VEMPs in PwMS

Thirteen of the 14 included studies that assessed VEMPs reported abnormalities in PwMS. Significant differences

between PwMS and controls were reported by 8 of the 10 studies that compared the two groups.[30 35 50 58 60–63] Both Kavasoğlu *et al* and Cochrane *et al* did not find differences between PwMS and controls.[59 65] However, Kavasoğlu *et al* found VEMP latency delays exceeding 2.5 SD from normative data in 23% of their PwMS[65]; Cochrane *et al* did not report whether any VEMP abnormalities were found in their PwMS.[59] Kavasoğlu *et al* included PwMS who were recently diagnosed (within less than 1 year), this may be the reason why they did not find differences between PwMS and controls.[65] Cochrane *et al* attributed the lack of difference between PwMS and controls to the fact that their participants had low scores on the disability scale indicating that they were less likely to have lesions affecting VEMP results.[59] However, Gabelić *et al*[58] found differences between PwMS and controls and their participants had similar disability scale scores to Cochrane *et al*.[58 59] The remaining studies that reported per cent PwMS who had abnormal VEMPs found various results. For example, Alpini *et al* found abnormalities in 70%.[29] Versino *et al* found abnormalities in 31.4%[28] and Koura and Hussein found abnormalities in 100% of PwMS who participated in their study.[66] The majority did not report type or duration of MS, or severity of impairment; therefore, it is difficult to speculate why some studies reported higher percentages of VEMP abnormalities in PwMS than others. The most common reported cVEMP abnormalities were prolonged p13 and n23 latencies[28–30 35 58 61–66] and the most common reported oVEMP abnormalities were prolonged p1 and n1 latencies.[35 58] Prolonged latencies may be a result of inflammatory demyelination in the vestibular nuclei and the root entry zone of the eighth cranial nerve[7] leading to reduced neural transmission speed and consequently prolonged latencies. Additionally, Aidar and Suzuki and Koura and Hussein noted the absence of waves in 30% and 40% of PwMS,[62 66] respectively, which may be caused by a lesion in the vestibulospinal tract.[76] Although all studies showed VEMP abnormalities in PwMS, the nature of these abnormalities and their relationships to MS could not be established. This is due to the differences between studies in VEMP stimulus and recording parameters (eg, presentation levels, stimulus polarity, transducer), resulting in the inability to combine data for a meta-analysis. In addition, type, duration and severity of MS were not consistently reported. Therefore, there is a clear need for consistency across studies to better understand the effects of MS on the vestibular system by using VEMPs.

### Rotary chair testing in PwMS

The two studies that reported results on rotary chair testing from the same participants found differences that indicate central rather than peripheral vestibular impairments in PwMS compared with controls.[59 67] They suggested that central integration of peripheral input is what is impaired in PwMS.[59 67] However, more studies are needed to confirm these findings.

### Clinical implications

This systematic review clearly demonstrates that both peripheral and central audio-vestibular functions may be affected by MS plaques and by the MS inflammatory process. This suggests that a comprehensive audio-vestibular test battery should be incorporated in the assessment protocols of PwMS. For auditory function, this review found that ABRs were the most sensitive measure in PwMS; therefore, ABRs should be included in the test battery. OAEs and PTA should also be included given that there is some evidence that the inner ear may be affected by the MS inflammatory process. For vestibular function, this review found that all tests are affected in PwMS; therefore, ENG/VNG, VEMPs, rotary chair and posturography should be included in the battery.

## CONCLUSIONS

This systematic review aimed at understanding the effects of MS on the auditory and vestibular systems. No previous review that investigated this relationship was found. In this systematic review, an effect of MS on the auditory system was consistently found in the results of ABRs, CAEPs, WDS and GIN tests, but not in OAEs or PTAs, indicating that the brainstem and central auditory systems were more likely to be affected by MS than the peripheral auditory system. Furthermore, an effect of MS on the vestibular system was consistently found in the results of ENG, dynamic and static posturography and VEMPs. This suggests that both the peripheral and central vestibular systems may be involved in the degenerative process of MS. However, quantitative analyses could not be performed in this systematic review due to inconsistencies across studies in reporting of results, in test procedures and in test stimulus and recording parameters. Therefore, although abnormal results in PwMS on both auditory and vestibular tests were identified, further studies are necessary to quantify this effect and obtain more robust evidence on the effects of MS on the auditory and vestibular systems. Future studies should use consistent test parameters and account for confounding factors such as type, duration and severity of MS.

**Contributors** EG and GB equally contributed to this work (cofirst authors). EG, GB and KK conceptualised, planned and designed this systematic review. EG conducted the searches and initial screening of titles and abstracts. EG and GB independently reviewed the full texts and conducted the assessment of bias. EG extracted the data and this was verified by GB, and disagreements at any stage were resolved by discussion and by involvement of KK. EG and GB interpreted the results and this was reviewed by KK. EG and GB wrote the original manuscript and also reviewed and edited the manuscript. KK was responsible for supervision, administration, reviewing, editing the manuscript, and is responsible for the overall content as the guarantor.

**Funding** KK was supported by the NIHR Manchester Biomedical Research Centre (UK IS-BRC-1215-20007) and MRC Programme Grant (MR/L003589/1).

**Competing interests** None declared.

**Patient and public involvement** Patients and/or the public were not involved in the design, or conduct, or reporting, or dissemination plans of this research.

**Patient consent for publication** Not applicable.

**Ethics approval** As new participant data were not collected, ethics approval was not needed.

**Provenance and peer review** Not commissioned; externally peer reviewed.

**Data availability statement** All data relevant to the study are included in the article or uploaded as supplementary information.

**ORCID iDs**
Ghada Binkhamis http://orcid.org/0000-0002-7696-1073
Karolina Kluk http://orcid.org/0000-0003-3638-2787

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
