## [Reviewer comments · BMJ Open]

ARTICLE DETAILS

TITLE (PROVISIONAL)	The effects of Multiple Sclerosis on the audio-vestibular system: A systematic review
AUTHORS	Gür, Evrim; Binkhamis, Ghada; Kluk, Karolina

VERSION 1 – REVIEW

REVIEWER	Pinero-Pinto, Elena Universidad de Sevilla, Departamento de Fisioterapia
REVIEW RETURNED	09-Jan-2022

GENERAL COMMENTS	Thank you for the opportunity to review the manuscript entitled "The effects of Multiple Sclerosis on the audio-vestibular system: A systematic review". The information presented by the authors is very interesting, as well as applicable to the health scientific community that cares for patients with MS. Here are some comments and suggestions to improve the quality of the manuscript: - In the abstract, it would be interesting to include the tools for assessing the quality of the articles used.- The introduction should be shortened and the content better clarified.- Lines 130-133 can be deleted.- Lines 142 to 145 must be included in the selection criteria, not before. Why are there no search limitations by language or by date if they are later excluded? It would be good to restructure this section.- In the description of the studies, in the results, always use the numbers written in the same way (lines 207-212, 223-226).- The discussion is very well structured, I would just remember that the "n" is usually in italics, as in figure 1.
--

REVIEWER	Hernández-Rodríguez, Juan-Carlos Virgen del Rocio University Hospital, Dermatology Department
REVIEW RETURNED	19-Jan-2022

GENERAL COMMENTS	Comments to the author: - The presented manuscript is a systematic review aimed to summarise the scientific literature available for audio-vestibular effects of Multiple sclerosis. No quantitative analysis was possible due to the heterogeneity from current literature. This review consider an interesting and booming topic. However, major changes have to be performed to improve methodological quality of the study. I have provided some comments and suggestions to improve the manuscript. ABSTRACT:
--

	 - I suggest that sections design, setting, eligibility criteria, participants and primary and secondary outcomes should be included in a unified section titled methods. This will help the lecturer to identify better the information from the abstract and follows PRISMA 2020 abstract checklist. - Page 3/46, line 9-10: The sentence “study selection and assessments of bias were independently conducted by two reviewers” is not an eligibility criteria. Therefore, it has to be removed from the abstract. INTRODUCTION:  - I am sorry to say that the introduction needs a major review to be considered for publication. The included information is reiterative and under my point of view, it is difficult to read for the lecturer. - It is necessary to write the name of the “INTRODUCTION” section. - First part of the introduction referred to pathogenesis is complicated to be related to audio-vestibular problems in MS. Please clarify this statement and provide more literature that supports the idea. - Page 5/46, line 45: change ; by (). (focal areas of demyelination) - Page 7/46, line 91,92: The statement “in addition, peripheral vestibular deficits may also coexist” is redundant with line 89. - Page 7/46, line 103: Please, explain the control group from Inojosa et al. It is unclear. - Last paragraph is a summary from previous paragraph. Also, the justification that literature must be summarised is repeated several times. I suggest that main ideas of the introduction have to be more direct and clear. METHODS:  - Specified search terms within the manuscript are not real search terms, but search strategies. A single search terms would be “Multiple sclerosis”. - As PRISMA 2020 statements reports all search strategies need to be reported from all databases. Please, provide this information, for example as supplementary material. - Also, number of records must be placed in results section, following the PRISMA statement - Page 9/46, line 142-145 exclusion criteria are specified, but it should be explained in the next section. - Exclusion and inclusion criteria are not opposites, for example, PwMS above 18 years old were included in the manuscript. Then you reported that PwMS under 18 years old were excluded. - A small summary of how outcomes are measured would help the lecturer to understand the aims of mentioned test battery. RESULTS:  - Tables from results are clear and provide interesting information. I congrats the authors for the great effort. - Only one suggestion for Table 1 and 2, it would have been interesting if EDSS score from included studies to find if a trend between EDSS scores and audio-vestibular problems in MS. For example, an EDSS above 6 points indicates postural problems, when vestibular problems are the main cause of imbalance. To know EDSS score from studies may help to understand this issue. - Division between outcomes of study should be better defined to ease lecture. - Please define in depth what control participants are included in the different studies. DISCUSSION:  - The discussion section would benefit from a specific paragraph of clinical implications from the reviewed results. Some aspects are
--	---

	explained in the conclusion section, but it will be interesting a wider explanation of clinical characteristics of PwMS and audio-vestibular outcomes.  - Please provide specific information when you talk about controls along the discussion text. Healthy? PwMS without auditory-vestibular disorders? - Pag 22/46, line 421-426 periheral and central aetiologies could be removed from the discussion and moved to the introduction. Central causes have been mention in the introduction section, but peripheral not.
--	---

VERSION 1 – AUTHOR RESPONSE

Reviewer: 1

Dr. Elena Pinero-Pinto, Universidad de Sevilla

Comments to the Author:

Thank you for the opportunity to review the manuscript entitled "The effects of Multiple Sclerosis on the audio-vestibular system: A systematic review". The information presented by the authors is very interesting, as well as applicable to the health scientific community that cares for patients with MS.

Here are some comments and suggestions to improve the quality of the manuscript:

Thank you for your comments and suggestions that have helped improve the quality of the manuscript.

- 1) In the abstract, it would be interesting to include the tools for assessing the quality of the articles used.
 - We have added these to the abstract
 - Lines 12 – 14 in the revised manuscript: “*Study selection and assessments of bias were independently conducted by two reviewers using the Risk of Bias Assessment Tool for Nonrandomized Studies, Newcastle Ottawa Scale (NOS), and the NOS adapted for cross-sectional studies.*”

- 2) The introduction should be shortened and the content better clarified.
 - We have shortened the introduction from 1089 words to 762 words, and better clarified its' contents.

- 3) Lines 130-133 can be deleted.
 - Unfortunately lines 130 – 133 (Lines 106 – 108 in the revised Manuscript) cannot be deleted as a statements on patient and public involvement and ethical approval are required by the journal.

- 4) Lines 142 to 145 must be included in the selection criteria, not before. Why are there no search limitations by language or by date if they are later excluded? It would be good to restructure this section.
 - We moved those lines under the “Inclusion and exclusion criteria” subheading and edited this section for better clarity (Lines 117 – 128 in the revised manuscript).
 - For clarity: we have edited the methods section and added a subheading for “Audio-vestibular tests” (Lines 129 – 140 in the revised manuscript) and another for “Study selection” (Lines 141 – 144 in the revised manuscript).

- In hindsight, yes it would have been better to include search limitations by language and by date in the original search. We will ensure to do this in the future.
- 5) In the description of the studies, in the results, always use the numbers written in the same way (lines 207-212, 223-226).
- We have used APA style – where words are used for numbers from zero to nine and numbers are used for 10 and above. Words are also used for numbers in the beginning of a sentence (<https://apastyle.apa.org/style-grammar-guidelines/numbers/words>).
- 6) The discussion is very well structured, I would just remember that the "n" is usually in italics, as in figure 1.
- Thank you.
 - We have changed all “n” to “*n*” (italics) in the discussion and in figure 1.

Reviewer: 2

Dr. Juan-Carlos Hernández-Rodríguez, Virgen del Rocio University Hospital
Comments to the Author:

- The presented manuscript is a systematic review aimed to summarise the scientific literature available for audio-vestibular effects of Multiple sclerosis. No quantitative analysis was possible due to the heterogeneity from current literature. This review consider an interesting and booming topic. However, major changes have to be performed to improve methodological quality of the study. I have provided some comments and suggestions to improve the manuscript.

Thank you for your comments and suggestions that have helped improve the quality of the manuscript.

ABSTRACT:

- 1) I suggest that sections design, setting, eligibility criteria, participants and primary and secondary outcomes should be included in a unified section titled methods. This will help the lecturer to identify better the information from the abstract and follows PRISMA 2020 abstract checklist.
- We have modified the abstract to have a “methods” section and removed the sections design, setting, eligibility criteria, participants and primary and secondary outcomes. (We have also incorporated comments on the abstract from Reviewer 1)
 - Lines 4 – 17 (in the original manuscript) now read (lines 4 – 14 in the revised manuscript): ***Methods:*** *Systematic review of literature investigating audio-vestibular conditions in persons with MS (PwMS) aged ≥ 18 years. PUBMED, SCOPUS, NICE, Web of Science were searched. Randomised control trials, cohort, case-control, observational, retrospective studies in English, published from 2000 to 21st November 2021, evaluated PwMS with at least one outcome (Pure Tone Audiometry, Auditory Brainstem Response, Otoacoustic Emissions, Cortical Auditory Evoked Potentials, Functional MRI assessing auditory function, Vestibular Evoked Myogenic Potentials, Videonystagmography, Electonystagmography, Posturography, Rotary Chair, Gaps-in-Noise, Word Discrimination Scores, Duration Pattern Sequence Test) were included. Study selection and assessments of bias were independently conducted by two reviewers using the Risk of Bias Assessment Tool for Nonrandomized Studies, Newcastle Ottawa Scale (NOS), and the NOS adapted for cross-sectional studies.”*

2) Page 3/46, line 9-10: The sentence “study selection and assessments of bias were independently conducted by two reviewers” is not an eligibility criteria. Therefore, it has to be removed from the abstract.

- Lines 9 – 10 (from the original manuscript) have been deleted from the abstract.

INTRODUCTION:

3) I am sorry to say that the introduction needs a major review to be considered for publication. The included information is reiterative and under my point of view, it is difficult to read for the lecturer.

- We have edited and shortened the introduction plus better clarified its' contents. (Also recommended by the Associate Editor and Reviewer 1)

4) It is necessary to write the name of the “INTRODUCTION” section.

- We have added the heading “INTRODCUTION” (Line 41 in the revised manuscript).

5) First part of the introduction referred to pathogenesis is complicated to be related to audio-vestibular problems in MS. Please clarify this statement and provide more literature that supports the idea.

- As part of the edits and the shortening of the introduction, we have simplified the background on MS and kept it relevant to the audio-vestibular system.

6) Page 5/46, line 45: change ; by (). (focal areas of demyelination)

- This has been changed, line 45 in the original manuscript now reads (line 46 in the revised manuscript): “... *CNS plaques (focal areas of demyelination)* ...”.

7) Page 7/46, line 91,92: The statement “in addition, peripheral vestibular deficits may also coexist” is redundant with line 89.

- This has been removed with the edits on the introduction

8) Page 7/46, line 103: Please, explain the control group from Inojosa et al. It is unclear.

- We have clarified that they were “healthy controls” and we have done this throughout the manuscript.

9) Last paragraph is a summary from previous paragraph. Also, the justification that literature must be summarised is repeated several times. I suggest that main ideas of the introduction have to be more direct and clear.

- We believe we have addressed this with the edits on the introduction.

METHODS:

10) Specified search terms within the manuscript are not real search terms, but search strategies. A single search terms would be “Multiple sclerosis”.

- We have modified this:
 - Lines 136 – 139 in the original manuscript now read (lines 112 – 113 in the revised manuscript): “... *following terms: “Multiple Sclerosis”, “Auditory Function”, “Vestibular Function”, “Hearing Loss”, “Dizziness”, “Vertigo”.*”

11) As PRISMA 2020 statements reports all search strategies need to be reported from all databases. Please, provide this information, for example as supplementary material.

- We have added the search strategy to the supplement: Section I: Literature Search Strategy (Supplement pages 1 – 4).

12) Also, number of records must be placed in results section, following the PRISMA statement

- We have removed this from the methods (lines 140 – 145 from the original manuscript have been deleted)

13) Page 9/46, line 142-145 exclusion criteria are specified, but it should be explained in the next section.

- We have removed this from the “data sources” section (lines 142 – 145 from the original manuscript have been deleted)

14) Exclusion and inclusion criteria are not opposites, for example, PwMS above 18 years old were included in the manuscript. Then you reported that PwMS under 18 years old were excluded.

- We have modified this
 - Lines 148 – 170 in the original manuscript now read (lines 117 – 128 in the revised manuscript): ***Inclusion and exclusion criteria.*** *Studies were eligible for inclusion if they met the following criteria: 1) investigated audio-vestibular conditions; 2) participants were diagnosed with any type of MS (RRMS, SPMS, PPMS, PRMS); 3) participants were aged 18 years and older because of maturational effects on outcomes of some audio-vestibular tests; 4) one or more of the pre-selected tests described below were used to evaluate auditory or vestibular function; 5) any of the following study types: randomised controlled trials, cohort, case-controlled, observational, or retrospective studies; 6) results from PwMS were either compared to a healthy control group or to normative data; 7) any geographical location. Studies were excluded based on the following: 1) did not include the following keywords “Multiple Sclerosis”, “dizziness”, “auditory function”, “vestibular”, “vertigo” or “hearing loss” in the title or abstract; 2) no available full text in English; 3) published before the year 2000; 4) participants were diagnosed with comorbid diseases.”*

15) A small summary of how outcomes are measured would help the lecturer to understand the aims of mentioned test battery.

- We have referred the reader to the supplement for clarification on measures of each test by adding the following statement (lines 139 – 140 in the revised manuscript): *“(Refer to supplement for details on which measures and parameters were extracted for each test).”*

RESULTS:

16) Tables from results are clear and provide interesting information. I congrats the authors for the great effort.

- Thank you

17) Only one suggestion for Table 1 and 2, it would have been interesting if EDSS score from included studies to find if a trend between EDSS scores and audio-vestibular problems in MS. For example, an EDSS above 6 points indicates postural problems, when vestibular problems are the main cause of imbalance. To know EDSS score from studies may help to understand this issue.

- This is a very interesting point indeed, thank you.
- Following the reviewer’s comments, we have scanned the papers for EDSS scores and identified that more than half of the 35 papers included in this systematic review did not report EDSS scores, 14 papers reported mean and SD, and 2 studies used EDSS as an inclusion criterion. Thus, we feel that adding such minimal EDSS data would not help us answer our research question: “whether persons with MS experience auditory and vestibular disorders at a higher rate than the general population”. Unfortunately, a detailed look into EDSS scores (reflecting the severity of MS) and discussion of their relation to audio-vestibular function is beyond the scope of this systematic review.

18) Division between outcomes of study should be better defined to ease lecture.

- We have created alternatives to Tables 1 and 2 displaying results based on outcome and placed them in the supplement (Table S4 and Table S7). We refer to them in the revised manuscript as follows:
 - Lines 197 – 198: “... (Table 1 – see Supplement Table S4 for an alternative table format summarizing findings based on outcome)”
 - Lines 252 – 253: “... (Table 2 – see Supplement Table S7 for an alternative table format summarizing findings based on outcome)”

19) Please define in depth what control participants are included in the different studies.

- We have clarified throughout the manuscript that all control participants were “healthy controls”.

DISCUSSION:

20) The discussion section would benefit from a specific paragraph of clinical implications from the reviewed results. Some aspects are explained in the conclusion section, but it will be interesting a wider explanation of clinical characteristics of PwMS and audio-vestibular outcomes.

- We added a paragraph on clinical implications after the conclusions:
 - Lines 486 – 497 in the revised manuscript: “**CLINICAL IMPLICATIONS.** *This systematic review clearly demonstrates that audio-vestibular function is affected in PwMS. This suggests that tests of audio-vestibular function should be incorporated in the assessment protocols of PwMS. A test battery approach that includes tests of peripheral and central function would be most appropriate to ensure all potential regions that may be affected by MS plaques are evaluated. This review found that ABRs are the most sensitive measure of auditory function in PwMS; therefore, at least ABR testing must be included as part of the assessment of PwMS. Additionally, OAEs and PTA should also be included given that there is some evidence that the MS inflammatory process could affect the inner ear. For vestibular function, this review found that all tests are affected in PwMS suggesting that the vestibular test battery should be comprehensive and incorporate ENG/VNG, VEMPs, Rotary Chair, and Posturography to ensure that the whole vestibular pathway is evaluated.*”

21) Please provide specific information when you talk about controls along the discussion text. Healthy? PwMS without auditory-vestibular disorders?

- We have clarified throughout the manuscript that all control participants were “healthy controls”.

22) Pag 22/46, line 421-426 peripheral and central aetiologies could be removed from the discussion and moved to the introduction. Central causes have been mention in the introduction section, but peripheral not.

- This has been removed from the discussion and kept in the introduction.

VERSION 2 – REVIEW

REVIEWER	Pinero-Pinto, Elena Universidad de Sevilla, Departamento de Fisioterapia
REVIEW RETURNED	27-Apr-2022

GENERAL COMMENTS	The quality of the manuscript has improved considerably, as the authors have made all suggested changes.
--

REVIEWER	Hernández-Rodríguez, Juan-Carlos Virgen del Rocio University Hospital, Dermatology Department
REVIEW RETURNED	22-May-2022

GENERAL COMMENTS	Comments to the authors: We appreciate the effort of the authors to answer all our queries. I have some changes to propose before the acceptance if editor will consider. Introduction: 1. Page 5/96 line 43: the word autoimmune is repeated 2 times respect to the word immune-mediated in definitions. In my opinion this could be redundant, and I suggest removing it.2. Page 5/96 line 44: the term “subtype” could be replaced by “phenotypes”. Results: 3. Table 1: in abbreviations, I do not find the abbreviation of NA. Please, provide it. Discussion: 4. The “clinical implications” should be placed before the conclusions.5. It would be interesting that the limitations of the review appeared after the clinical implications. Since my point of view, this would help the reader because it follows the PRISMA statement 2020.6. Page 26/96 line 490: Please, change the word “plagues” by “plaques”.7. I appreciate the effort of adding the section “clinical implications”, but it should be summarised. For instance, the idea “the need of vestibular assessment in PwMS” appeared at least three times.
--

VERSION 2 – AUTHOR RESPONSE

Reviewer: 1

Dr. Elena Pinero-Pinto, Universidad de Sevilla

Comments to the Author:

The quality of the manuscript has improved considerably, as the authors have made all suggested changes.

Thank you for your positive feedback.

Reviewer: 2

Dr. Juan-Carlos Hernández-Rodríguez, Virgen del Rocio University Hospital

Comments to the Author:

We appreciate the effort of the authors to answer all our queries. I have some changes to propose before the acceptance if editor will consider.

Thank you for your comments and suggestions that have helped further improve the quality of the manuscript.

INTRODUCTION:

1. Page 5/96 line 43: the word autoimmune is repeated 2 times respect to the word immune-mediated in definitions. In my opinion this could be redundant, and I suggest removing it.
 - We have deleted this.

2. Page 5/96 line 44: the term “subtype” could be replaced by “phenotypes”.
 - Thank you for this comment, we prefer to use the term subtypes or types as the term phenotypes is more commonly used in reference to genetic diseases or genotypes.
 - We replaced subtypes with types and have modified the sentence to incorporate comment number 1. The sentence now reads:
 - Lines 43 – 44 (in the original manuscript) now read (lines 43 – 44 in the revised manuscript): *“There are four main types of MS: ...”*

RESULTS:

3. Table 1: in abbreviations, I do not find the abbreviation of NA. Please, provide it.
 - Thank you for pointing this out, we have added this to the list of abbreviations.

DISCUSSION:

4. The “clinical implications” should be placed before the conclusions.
 - We have moved the clinical implications section as suggested.

5. It would be interesting that the limitations of the review appeared after the clinical implications. Since my point of view, this would help the reader because it follows the PRISMA statement 2020.
 - Thank you for your suggestion, we have placed the limitations early in the discussion as per this journal’s recommended structure.

6. Page 26/96 line 490: Please, change the word “plagues” by “plaques”.
 - Thank you for pointing this out, we have corrected this.

7. I appreciate the effort of adding the section “clinical implications”, but it should be summarised. For instance, the idea “the need of vestibular assessment in PwMS” appeared at least three times.
 - We have summarised this as follows:
 - Lines 487 – 497 (in the original manuscript) now read (lines 470 – 478 in the revised manuscript): *“This systematic review clearly demonstrates that both peripheral and central audio-vestibular functions may be affected by MS plaques and by the MS inflammatory process. This suggests that a comprehensive audio-vestibular test battery should be incorporated in the assessment protocols of PwMS. For auditory function, this review found that ABRs were the most sensitive measure in PwMS; therefore, ABRs should be included in the test battery. OAEs and PTA should also be included given that there is some evidence that the inner ear may be affected by the MS inflammatory process. For vestibular function, this review found that all tests are affected in PwMS; therefore, ENG/VNG, VEMPs, Rotary Chair, and Posturography should be included in the battery.”*

VERSION 3 – REVIEW

REVIEWER	Hernández-Rodríguez, Juan-Carlos Virgen del Rocio University Hospital, Dermatology Department
REVIEW RETURNED	26-Jun-2022

GENERAL COMMENTS	The authors made the effort to accomplish all the queries of reviewers. Thanks.
---